# The Variability of Snow Cover and Its Contribution to Water Resources in the Chinese Altai Mountains from 2000 to 2022

**Fengchen Yu** [1,2]**, Puyu Wang** [1,2,*]**, Lin Liu** [2]**, Hongliang Li** [1] **and Zhengyong Zhang** [2]

[1] State Key Laboratory of Cryosphere Science, Northwest Institute of Eco-Environment and Resources, Chinese Academy of Sciences, Lanzhou 730000, China; yufengchen@stu.shzu.edu.cn (F.Y.); lihongliang@nieer.ac.cn (H.L.)

[2] College of Sciences, Shihezi University, Shihezi 832000, China; llin@shzu.edu.cn (L.L.); zyz0815@shzu.edu.cn (Z.Z.)

[*] Correspondence: wangpuyu@lzb.ac.cn

**Abstract:** As one of the major water supply systems for inland rivers, especially in arid and semi-arid regions, snow cover strongly affects hydrological cycles. In this study, remote sensing datasets combined with in-situ observation data from a route survey of snow cover were used to investigate the changes in snow cover parameters on the Chinese Altai Mountains from 2000 to 2022, and the responses of snow cover to climate and hydrology were also discussed. The annual snow cover frequency (SCF), snow cover area, snow depth (SD), and snow density were 45.03%, $2.27 \times 10^4$ km$^2$, 23.4 cm, and ~0.21 g·cm$^{-3}$, respectively. The snow water equivalent ranged from 0.58 km$^3$ to 1.49 km$^3$, with an average of 1.12 km$^3$. Higher and lower SCF were mainly distributed at high elevations and on both sides of the Irtysh river. The maximum and minimum snow cover parameters occurred in the Burqin River Basin and the Lhaster River Basin. In years with high SCF, abnormal westerly airflow was favorable for water vapor transport to the Chinese Altai Mountains, resulting in strong snowfall, and vice versa in years with low SCF. There were significant seasonal differences in the impact of temperature and precipitation on regional SCF changes. The snowmelt runoff ratios were 11.2%, 25.30%, 8.04%, 30.22%, and 11.56% in the Irtysh, Kayit, Haba, Kelan, and Burqin River Basins. Snow meltwater has made a significant contribution to the hydrology of the Chinese Altai Mountains.

**Keywords:** Chinese Altai Mountains; snow cover frequency; snow depth; snow water equivalent; atmospheric circulation; water resources

## 1. Introduction

Snow cover is highly sensitive to subtle climate changes and, as one of the extremely active environmental elements on the ground surface, it is not only widely distributed but also exhibits significant seasonal variations [1,2]. As an important part of the cryosphere, snow cover strongly affects the energy budget and hydrological cycle between the land and atmosphere based on its high albedo and physical properties [3,4]. The discussion of snow cover changes driven by large-scale atmospheric circulation is helpful for understanding the response of snow cover physical change mechanisms to climate parameters [2,5]. Against the backdrop of current climate warming, snow cover parameters exhibit different characteristics. From 1972 to 2022, the snow cover area in the Northern Hemisphere continued to decrease at a rate of $0.14 \times 10^6$ km$^2$ (10a)$^{-1}$ [6]. The reduction of snow cover will unavoidably result in a decrease in the important water resources of snowmelt. Snow cover plays an indispensable role in ecological management, disaster prevention and control, and regional economic development [7–9]. Snow meltwater is not only an important freshwater resource for human production and life but also an important source of river runoff, especially in arid and semi-arid areas [10].

Ground station observations can provide high-quality snow cover information over a long period of time, providing strong data support for evaluating snow cover changes, but

their spatial representativeness is poor [11,12]. Remote sensing data offer the advantages of broad coverage, frequent updating, and high spatial resolution, which compensate for the limitations of ground-based monitoring data [13,14]. Moderate-resolution imaging spectroradiometer (MODIS) snow products have emerged as the mainstream data of remote sensing snow cover products due to their high spatiotemporal resolutions. Additionally, MODIS snow products have been widely adopted in the study of snow cover changes in mountainous areas on regional or global scales [15–17]. The combination of remote sensing images and model simulations with in-situ observations is considered the best validation for regional investigations of snow cover [18,19]. However, it cannot be denied that clouds are the main influencing factor affecting the recognition of snow information by MODIS snow products [11,20]. When applying MODIS snow products for regional snow cover spatiotemporal research, attention should be paid to the accuracy of MODIS snow products in identifying snow under the influence of cloud cover [15,21,22]. Previous studies have demonstrated the significant efficiency of the filtering techniques in cloud reduction, and the generated MOD10A2 snow products have very high consistency with ground snow cover observations [16,23–26]. Snow cover frequency (SCF) describes the cumulative frequency of snow cover during a specified period on a unit pixel scale, obtained by the ratio of the cumulative frequency of a single pixel within a period to the total number of statistics. By using remote sensing technology to obtain SCF from MOD10A2 products, it is possible to clearly understand the regional snow cover situation and reflect the overall characteristics of snow cover duration during a specified period.

The Chinese Altai Mountains are located in northern Xinjiang (Figure 1), which is accompanied by extremely rich seasonal snow meltwater resources, and the snowmelt runoff ratios are more than 50% in some basins [8,27]. Snow meltwater accounts for 15–25% of total river runoff in China's three major stable snow regions [28–30]. Previous studies show that since the 1960s, due to the impact of climate change, the date of snowmelt in northern Xinjiang has been advanced by 20–30 days, with a significant decrease in snow cover days and snow duration days but, conversely, a significant increase in snow depth (SD) [31–33]. For the Chinese Altai Mountains, current research only shows that the snow meltwater in the Kayit River Basin accounts for 29.3% of the annual runoff volume, and the Kelan River Basin accounts for 45% of the annual runoff volume, with even more than 60% in the snow melt season [8,30,34,35]. In addition, Zhong et al. (2021) [33] discussed the impacts of landscape and climatic factors on snow cover parameters. However, most previous studies focus on a single snow parameter of a specific River Basin or a specific route, lacking a comprehensive survey of the snow parameters of the Chinese Altai Mountains, particularly with a large uncertainty in the estimation of SD and snow water equivalent (SWE).

Against this background, snow cover variations in the Chinese Altai Mountains from 2000 to 2022 were investigated based on MODIS and in-situ observations, and the impact of climate factors on snow cover parameters was analyzed. The contribution of snow cover to regional water resources was also discussed. The specific purposes of this study were (1) to quantify the spatiotemporal variability of snow cover in the Chinese Altai Mountains in detail through various snow cover parameters, such as SCF, SD, snow density, and SWE; (2) analyze the physical mechanism of climate factors driving SCF changes; and (3) elucidate the contribution of snow meltwater to the annual runoff in each basin of the Chinese Altai Mountains. This study can provide a scientific reference for formulating the rational utilization and regional planning of snow cover resources in the Chinese Altai Mountains.

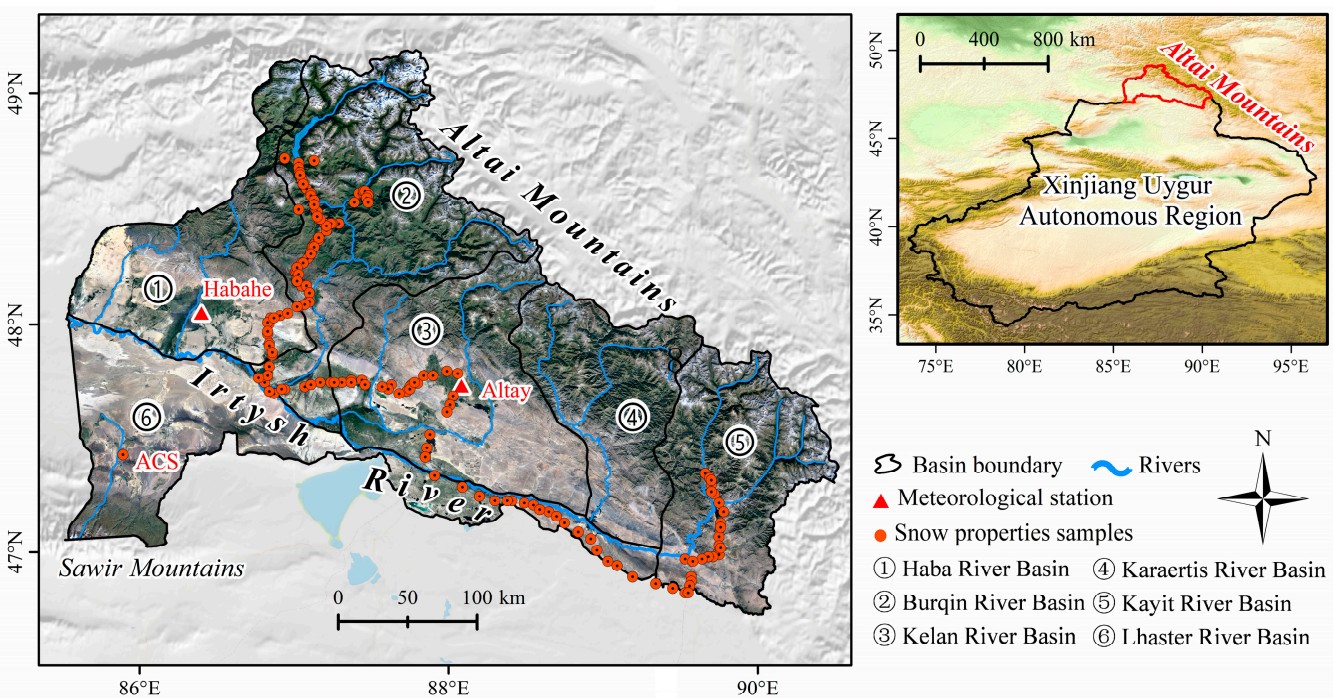

**Figure 1.** Location and extent of Chinese Altai Mountains. Snow properties samples are provided by Zhong et al. (2021) [33].

## 2. Study Area

The Altai Mountains are an international mountain range in central Asia, with a northwest–southeast trend, located at the junction of Russia, China, Mongolia, and Kazakhstan, stretching more than 2000 km from east to west. The highest peak, Friendship Peak (4373 m a.s.l.), is located at the junction of China and Mongolia at the source of the Kanas River in the upper reaches of the Burqin River. Friendship Peak, Kuitun Peak (4104 m a.s.l.), and several peaks above 4000 m a.s.l. constitute the largest modern glacier distribution center in the Altai Mountains.

The Chinese Altai Mountains are located in the Northern Xinjiang Uygur Autonomous Region, which has a temperate continental climate (Figure 1). In summer, it is mainly supplied by westerly circulation and precipitation is abundant. Approximately 70% of the precipitation occurs from June to August, and the precipitation gradually decreases from northwest to southeast [36,37]. Winter generally lasts five to six months and is accompanied by abundant snowfall. Arctic air masses seep into the Chinese Altai Mountains along the Irtysh Valley, resulting in low temperatures and snowfall [38]. The runoff in the mountainous area converges into the Irtysh and Ulungur Rivers. The Irtysh River Basin is formed by the Haba, Burqin, Kelan, Karaertis, Kayit, and Lhaster River Basins. The Sawir Mountains are a transitional section between the Tian Shan and central Altai Mountains, located in the Lhaster River Basin. The river water supply mainly depends on snow, glacier meltwater, and atmospheric precipitation, which have obvious seasonal variation characteristics [39]. However, seasonal snowmelt is the main water resource for river supply in the region, rather than glacier meltwater and atmospheric precipitation [27,29,34]. Based on Shen et al. (2007) [8] and Li et al. (2018) [40], since the 1950s, air temperatures and precipitation in the Chinese Altai Mountains have increased at rates of 0.8 °C·(10a)$^{-1}$ and 4.32 mm·(10a)$^{-1}$, respectively, with an increasing trend in maximum SD. With the increase in snow cover and air temperature in winter, the proportion of snowmelt runoff in the annual river runoff also increases.

## 3. Data and Methods

### 3.1. Data

#### 3.1.1. MODIS Snow Cover Products

The 8-day composite MODIS products (MOD10A2-v006) from 2000 to 2022 were obtained to investigate the spatiotemporal variability of SCF, including two MODIS tiles (tile references: h23v04 and h24v04), in this study. The spatial resolution of the MODIS/Terra 8 daily emerging snow products of 500 m was provided by the National Snow and Ice Data Center (NSID, http://nsidc.org/, accessed on 18 May 2023). The temporal extent for the 8-day composite MODIS/Terra snow cover products was from 24 February 2000 to 27 December 2022.

Reorganized this part: The 8-day composite MODIS products (MOD10A2-v006) with a spatial resolution of 500 m, were applied to inquire into the spatiotemporal variability in SCF. The two tiles with the track numbers of h23v04 and h24v04 cover the entire Chinese Altai Mountains, generated by the National Snow and Ice Data Center NSID, http://nsidc.org/, accessed on 18 May 2023). The temporal extent for the 8-day composite MODIS/Terra snow cover products range from 24 February 2000 to 27 December 2022.

#### 3.1.2. Meteorological Reanalysis Data

Reanalysis products have been widely used in studies on global climate change. The reanalysis dataset used in this study consisted of two main types: (1) Compared with the early global reanalysis, the ERA5 data generated by the European Center for Medium-Range Weather Forecasts (ECMWF https://cds.climate.copernicus.eu/, accessed on 20 May 2023), with higher temporal and spatial resolution, can capture atmospheric phenomena in more detail and is suitable for studying large-scale variability of the Asian abnormal atmospheric circulation model. In this study, ERA5 data were selected to analyze the relationship between the atmospheric circulation mechanism and SCF variability in the Chinese Altai Mountains. (2) The monthly air temperature and precipitation reanalysis data (https://data.tpdc.ac.cn/home, accessed on 20 May 2023) were generated from two types of climate datasets (global 0.5° and global high-resolution) using spatial downscaling methods, and their reliability was verified by in situ observations from 496 meteorological stations. The map projection of the set is a UTM Projection in the WGS 1984 coordinate system, with a spatial resolution of 0.0083333° (~1 km). The feedback of atmospheric circulation patterns on regional SCF variability occurs mainly through changes in meteorological parameters. Air temperature and precipitation are the most intuitive indicators of climate and atmospheric changes. Therefore, this dataset was used to investigate the response mechanisms of SCF to air temperature and precipitation changes under different modes in detail.

#### 3.1.3. Snow Depth and Snow Density

Snow depth and snow density are mainly used to estimate the snow water equivalent of the Chinese Altai Mountains. The specific information of each data is as follows:

(1)  Snow depth. A long-term series of daily snow depth derived from passive microwave remote sensing data, covering the entire Chinese Altai Mountains, was downloaded for free from the National Tibetan Plateau Data Center (TPDC https://data.tpdc.ac.cn/home, accessed on 20 May 2023), including daily snow depth estimates from 2000 to 2019, with a spatial resolution of 0.25°. The snow depth was generated by Che et al. (2021) [41] using a novel data fusion framework based on random forest regression combined with multisource snow depth product data, such as AMSR-E, AMSR2, NHSD, GlobSnow, ERA-Interim and MERRA2, geolocation (latitude and longitude), and topographic data (elevation). About 43,340 ground observation sites were used as the dependent variable to train and validate the snow depth, with a determination coefficient R2 of 0.81, root mean squared error of 7.69 cm, and mean absolute error of 2.74 cm. This dataset has been widely applied to regional climate and hydrological research [42–44]. Detailed information on the dataset sources and

product processes can be found in Hu et al. (2021) [45] or Hu et al. (2023) [19]. Additionally, we used in-situ observations of snow depth from 2017 to 2019 from the Altay and Habahe meteorological stations provided by the China Meteorological Administration (http://data.cma.cn/, accessed on 20 May 2023).

(2) Snow density. Zhong et al. (2021) [33] obtained 212 snow density data from the Chinese Altai Mountains in each January from 2015 to 2017 using a snow sampling tube (snow tubes). The study area consists of a mountainous higher-elevation region and the lower-elevation Irtysh River valley in the north, plus a small part occupied by the Sawir Mountains in the southwest corner (Figure 1). However, the snow density data sourced from Zhong et al. (2021) [33] and meteorological stations do not cover the Sawir Mountains. Thus, we obtained the snow density of Jimnai County from 2019 to 2021, located in the Sawir Mountains, with the support of the Altai Observation and Research Station of Cryospheric Science and Sustainable Development Northwest Institute of Eco-Environment and Resources (ACS) (Figure 1). Combined with more snow samples, it more reasonably reflects the snow density change of the Chinese Altai Mountains. Finally, combined with snow depth and snow density, the snow water equivalent was estimated based on the equation SWE = SD × Snow density × Snow cover area.

### 3.2. Methods

#### 3.2.1. Snow Cover Frequency Calculation

Using the MODIS reprojection tool (MRT) (Dwyer and Schmidt 2006), we set the two images as Universal Transverse Mercator projection (UTM) in the World Geodetic System-1984 Coordinate System (WGS-84), followed by format conversion (TIFF) and image mosaics.

Snow cover frequency was calculated to investigate the spatiotemporal changes of snow cover in the Chinese Altai Mountains. Two MOD10A2 tiles (tile references: h23v04 and h24v04) were set to a universal transverse Mercator projection (UTM) in the World Geodetic System-1984 Coordinate System (WGS-84), format conversion (TIFF), with image mosaics using MODIS reprojection tools (MRT) (Dwyer and Schmidt 2006). Assuming that regions with a pixel value of 1 are identified as snow cover and an area with a pixel value of 0 is identified as another land use type, pixel values of 100 (lake ice) and 200 (snow) within the Chinese Altai Mountains were converted to 1, and the other pixel values, such as 0 (missing data), 1 (no decision), 11 (night), 25 (no snow), 37 (lake), 39 (ocean), 50 (cloud), 254 (detector saturated), and 255 (fill), were converted to 0 through the reclassification method. SCF refers to the frequency of snow cover in a single snow cover pixel. The cell statistical tool in ArcGIS v.10.2 software was used to superimpose all the identified snow cover pixels in a specific year to obtain the annual SCF. The snow cover area during a specific period could also be extracted by calculating the number osnow-covereded pixels in each image.

#### 3.2.2. Analysis Methods

Linear regression analysis is one of the most commonly used methods for simulating the trend of a variable using a time series. Unified linear regression was used in this study to determine the quantitative relationship between SCF and time, which was applied to explore the magnitude of the trend and its statistical significance in SCF over time [5,46].

$$slope = \frac{n\sum_{i=1}^{n} SCF_i \times T_i - \sum_{i=1}^{n} SCF_i \times \sum_{i=1}^{n} T_i}{n\sum_{i=1}^{n} T_i^2 - \left(\sum_{i=1}^{n} T_i\right)^2} \tag{1}$$

where *slope* is the slope of the univariate linear regression equation, $SCF_i$ is the annual snow cover frequency in a specific (*i*) year, $T_i$ is a time variable, and *n* is the total number of years. In this study, *n* = 23. *slope* < 0 and *slope* > 0 indicate decreasing and increasing tendencies of annual SCF during the study period, respectively.

Significance levels (*p*) were used to describe the magnitude of the likelihood of sampling error, causing differences in the fit of the equations. The significance level of the SCF variation was determined using the F-test. The Pearson correlation coefficient was used for correlation analysis between snow and climate factors, including precipitation and air temperature [47]. Therefore, the climate-driven snow-cover change patterns during specific periods were further investigated in this study.

### 3.2.3. Accuracy Assessment

The uncertainty of the snow cover investigation mainly stemmed from clouds and data sources. The snow cover mapping algorithm applied to the MODIS data used spectral reflectance data from both visible and near-infrared wavelengths. Typically, the accuracy of snow cover recognition generally decreases with increasing clouds due to the high reflectivity of clouds in visible and near-infrared wavelengths [11,20]. Huang et al. (2007) [24] mentioned that under conditions of cloud cover greater than 20%, the snow cover recognition rate of MOD10A1 daily snow products decreased by 15% in northern Xinjiang. The weakening process of MODIS snow products includes data fusion (Terra and Aqua), spatial filters, and temporal filters [14,15]. The 8-day composite snow products effectively reduce the impact of clouds on snow cover. Therefore, based on the existing observation data and previous studies [16,23–26], we conducted a snow investigation using 8-day composite MOD10A2 snow products. MOD10A2 provides the maximum snow extent from MOD10A1 over a compositing period of 8 days on the same grid. The snow cover identification accuracy of MOD1OA2 improved by at least 29% in the Chinese Altai Mountains compared to MOD10A1 daily snow products [24]. Cloud obscuration is the main limitation of the MODIS snow cover product. Again, cloud coverage depends on region and season, but, most of the time, it is a practical issue [11,21,22]. Therefore, the specific impact of clouds on snow cover is hard to quantify. In the future, we will combine MODIS data in time and space to reduce the cloud coverage of map snow cover. In the current research, following the evaluation method proposed by Huang et al. (2007) [24] and Parajka and Blöschl 2008 [15], in-situ measurements from meteorological stations (Haba, Altai, and Jimnai, see Section 3.1.3) from 2018 to 2021 were selected to validate the accuracy of MOD10A2 using accuracy evaluation indicators such as total recognition accuracy, snow recognition accuracy, multi-score error, missed score error, and kappa. The total recognition accuracy was calculated by the ratio of the number of pixels that matched the actual and pixel values to the total number of pixels. Accurate snow cover identification meant that snow cover records within 8 days existed in both meteorological stations and MODIS images. Only snow cover records from meteorological stations were considered as missed points and only snow cover records in an image were considered as multiple points. Similar to the report in northern Xinjiang by Huang et al. (2007) [24], the total recognition accuracy for the Chinese Altai Mountains from 2018 to 2021 ranged from 78.3% to 84.8%, with an average of 81.5%. The snow cover identification accuracy retrieved from MOD1OA2 ranged from 73.5% to 83.8%, with an average of 78.8%. The kappa of 0.44 indicated that the snow cover identification accuracy from MOD10A2 was relatively consistent with the in-situ measurements. We believe that the partial difference between MOD10A2 and the in-situ measurements was caused by imbalanced training samples, which can be expected. The average multi-score error of 9.5% and missed score error of 21.1% were also acceptable in this region, which is consistent with previous research [24,26]. The results show that the MOD10A2 product can be applied for snow monitoring in the Chinese Altai Mountains.

## 4. Results

### 4.1. Spatial Pattern in SCF

The average SCF value in the Chinese Altai Mountains from 2000 to 2022 was approximately 45.03%, with significant spatial differences. The SCF values showed an increasing trend with altitude, extending northward from the Irtysh River to Friendship Peak and southward to the Sawir Mountains. The spatial pattern showed that the SCF values were

relatively low along rivers and plains and relatively high in glacier distribution areas and high-altitude areas (Figure 2). The regions with the highest SCF values (>80%) were principally in the northern Burqin River Basin and the southern Lhaster River Basin. A part of the region with SCF values of 40–80% was distributed in the southern part of the Lhaster River Basin, and the other part extended from the northern part of the Haba River Basin to the northern part of the Kayit River Basin, showing a northwest–southeast direction. The lowest SCF values of <20% were mainly concentrated at the junction of the Haba River Basin and Lhaster River Basin; areas with SCF values of 20–40% were mainly concentrated in the northern Lhaster River Basin and the other southern river basins, namely, those on both sides of the Irtysh River. SCF values were generally higher than 50% for the Burqin, Kayit, and Karaertis River Basins, with average SCF values of 58.35%, 54.9%, and 50.96%, respectively. The SCF values were generally lower than 40% for the Kelan, Haba, and Lhaster River Basins, with average SCF values of 39.16%, 36.86%, and 31.03%, respectively.

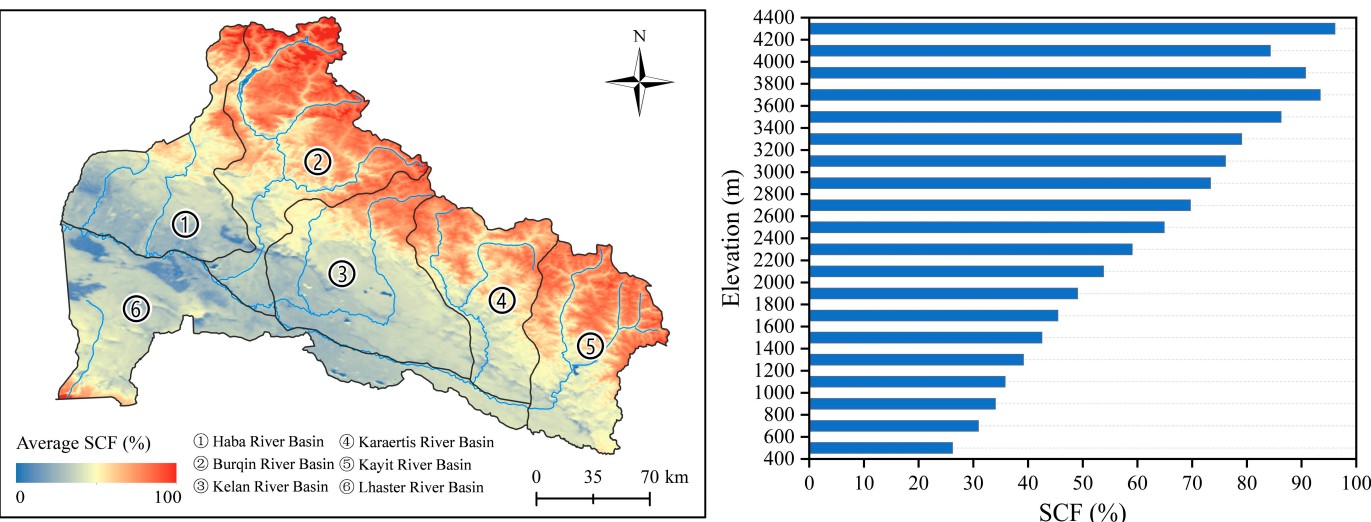

**Figure 2.** Spatial distribution of annual SCF in the Chinese Altai Mountains from 2000 to 2022.

The average SCF was higher than 50% in the Burqin (58.35%), Kayit (54.9%), and Karaertis (50.96%) river basins, but lower than 40% in the Kelan (39.16%), Haba (36.86), and Lhaster (31.03%) river basins.

*4.2. Interannual Variation in SCF*

As shown in Figure 3a, the average SCF value indicated a general increasing trend in the Chinese Altai Mountains from 2000 to 2022, with a growth rate of approximately 1.4% per 10 years. The regions in which SCF decreased were mainly along the northern part of the Irtysh River, accounting for 18% of the territory of the Chinese Altai Mountains. In total, 82% of all territory increase was mainly concentrated in the southern Lhaster River Basin and the northern river basins. There was obvious heterogeneity in the spatial and temporal patterns in the SCF, which was generally presented as a spatially northwest–southeast parallel line. Figure 3b illustrates that the regions with significant ($p < 0.05$) and extremely significant ($p < 0.01$) decreases in SCF were mainly in the southern part of the Kelan River Basin, accounting for less than 1% of the total area. The regions with extremely significant ($p < 0.01$) increases in SCF accounted for 0.82% of the total area, which was scattered in the Haba River Basin, Burqin River Basin, Lhaster River Basin, and Karaertis River Basin. In total, 4.94% of all territories increased significantly ($p < 0.05$), mainly distributed along the tributaries of the rivers. Overall, the regions with significant variations in SCF from 2000 to 2022 were spread along the river.

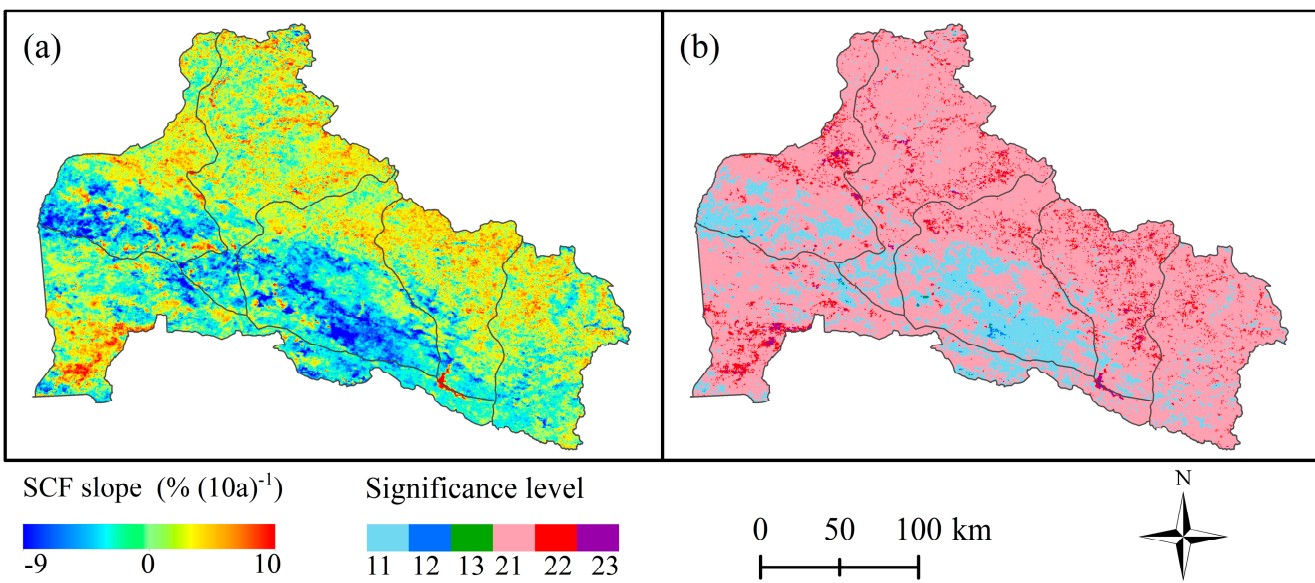

**Figure 3.** The slope of interannual changes in SCF (**a**) and the significant correlation with statistical significance (**b**) in the Chinese Altai Mountains from 2000 to 2022. In total, 11 represent a not statistically significant decreasing trend, 12 represent a decreasing trend at the 95% confidence level, 13 represent a decreasing trend at the 99% confidence level, 21 represent a not statistically significant increasing trend, 22 represent an increasing trend at the 95% confidence level, and 23 represent an increasing trend at the 99% confidence level.

*4.3. Seasonal Variation in SCF*

There were significant seasonal differences from 2000 to 2022 in the SCF variations, as shown in Figures 4 and 5. The four seasons were defined in this study as follows: spring, March to May; summer, June to August; autumn, September to November; and winter, December to February. The spatial pattern of SCF in spring and autumn was consistent with that of annual SCF (Figure 2), with average SCF values of 44.9% and 39.35%, respectively. The maximum SCF occurred in the Burqin River Basin, followed by the Kayit and Karaertis River Basins, while the Haba and Kelan River Basins were basically equal, and the minimum occurred in the Lhaster River Basin (Figure 5b). In spring, a downward trend in average SCF values was found in all basins, with a decrease rate of approximately 1.92% per decade and a decrease area of 77.07% in the Chinese Altai mountains (Figure 5). Regions with significant changes ($p < 0.01$ and $p < 0.05$) comprised approximately 2.73% of the territory of the Chinese Altai Mountains. There was an overall increasing trend in autumn, with an increasing rate of $1.23\% \cdot (10a)^{-1}$. The increased area accounted for 66% of the total area, with 2.82% showing a significant growth trend. Only approximately 0.2% of all territories showed a marked decreasing trend. The phenomenon of snow melting was gradually presented owing to the decrease in snowfall and the increase in air temperature in summer. The average SCF of the Chinese Altai Mountains was 4%, and the SCF of all basins was less than 3%, except for the Burqin (11.12%) and Kayit River Basins (5.22%). Approximately 32.96% of the total area was not covered by snow; the pixel value for these regions was noisy, as shown in Figure 4f. The average SCF value in winter was the highest (90%) among the four seasons. Only 1.20% of the total area had SCF values lower than 40%, which were primarily in the upper reaches of the Irtysh River. In winter, SCF variations exhibited an increasing trend, with an increase rate of $5.14\% \cdot (10a)^{-1}$, accounting for 94.11% of the total area. The regions with significant increases accounted for 5.76% of the total territory, primarily in the Haba, Burqin, and Kelan River Basins. The trend of SCF showed a significant cyclical pattern, with an overall increase from spring to winter. There was a positive gradient in autumn and winter but it returned to a negative gradient in spring and remained stable in summer (Figure 5a). The standard deviation of SCF value in spring

was the largest, reaching 31.75%, indicating that the regional snow cover difference was the strongest in spring. In general, although the spatiotemporal variability of SCF in a single season variable showed a relatively weak trend, there were significant differences in SCF changes among the four seasons, which also reflects the susceptibility of SCF to climate change.

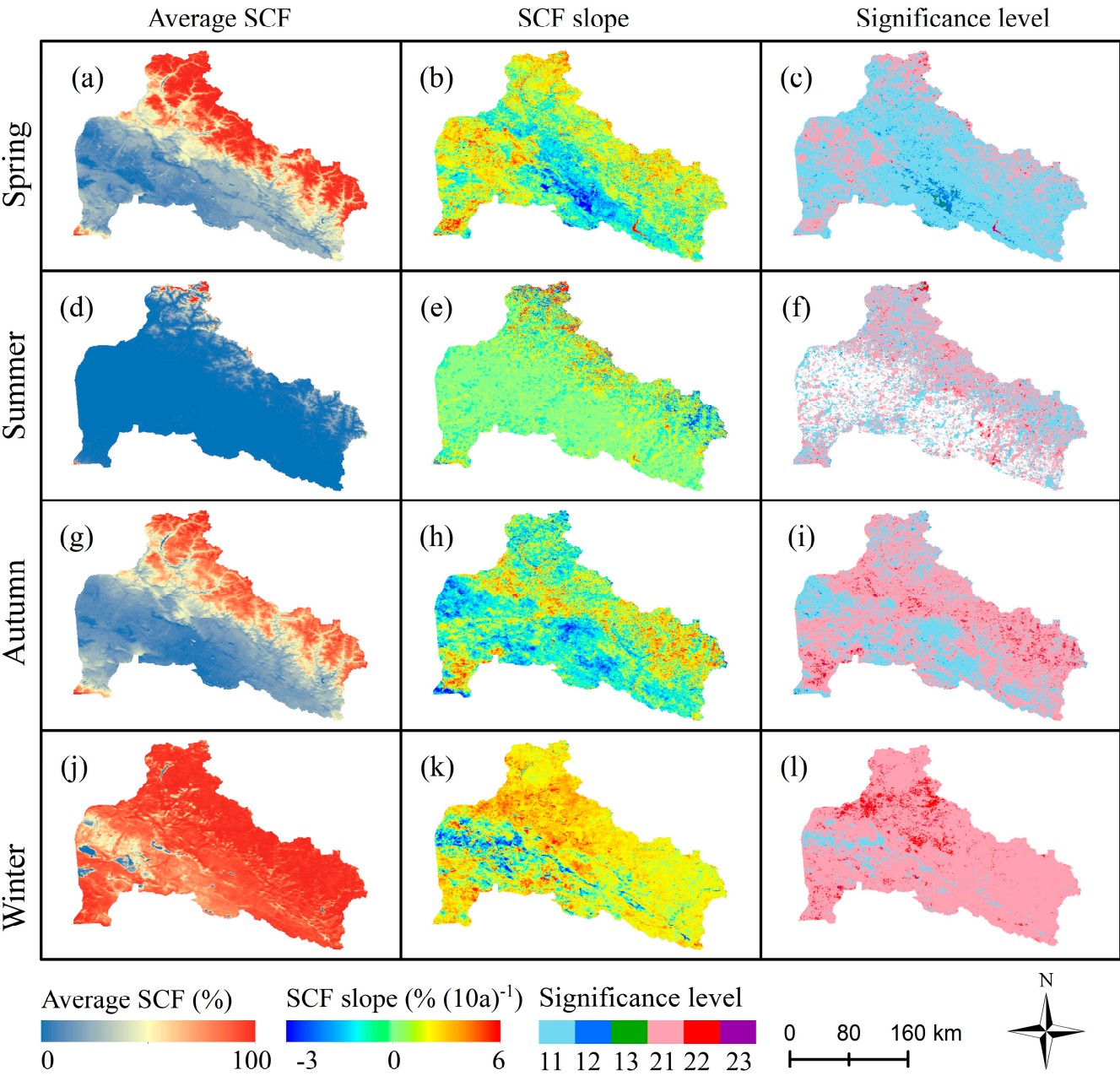

**Figure 4.** Temporal variation and statistically significant variations of SCF in the Chinese Altai Mountains based on the image metric scale in different seasons. Among them, the spatial pattern (**a**,**d**,**g**,**j**), unit pixel change rate (**b**,**e**,**h**,**k**), and the significant correlation with statistical significance (**c**,**f**,**i**,**l**) in the four seasons was shown.

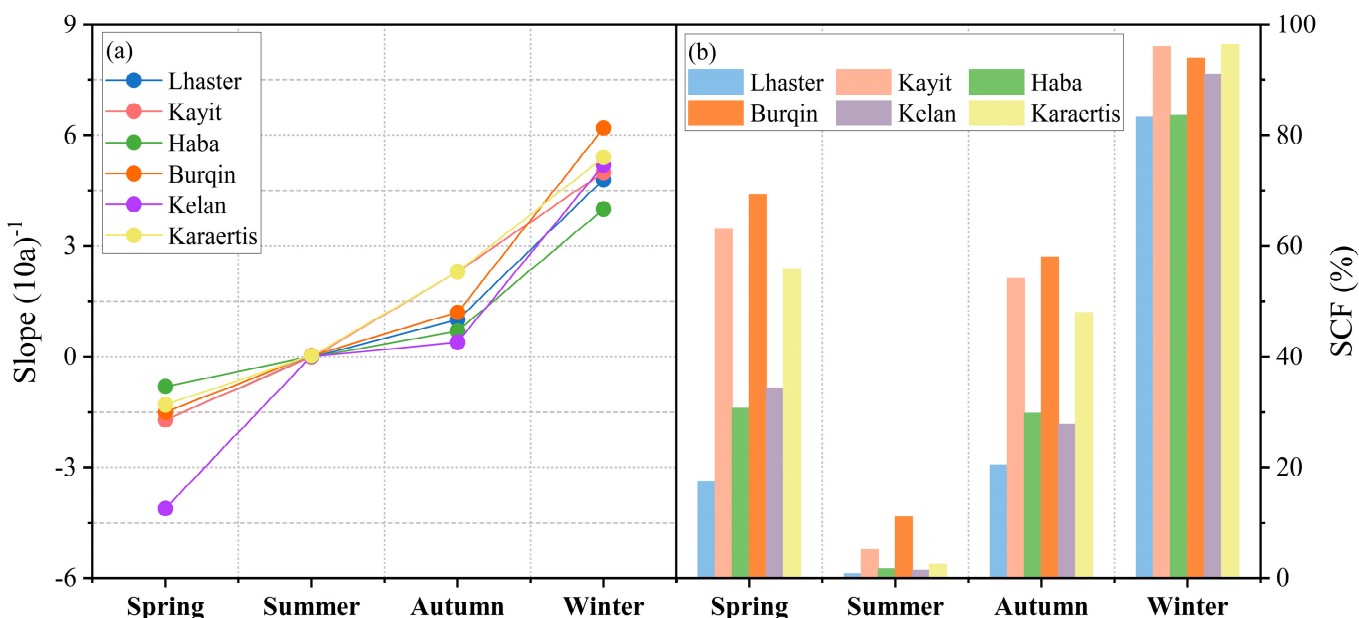

**Figure 5.** Trend (**a**) and SCF (**b**) variations in the basins of the Chinese Altai Mountains in different seasons.

*4.4. Snow Depth, Snow Density, and Snow Water Equivalent*

According to the daily snow depth dataset from Che et al. (2021) [41], based on machine learning methods, the snow depth was greater in higher-elevation mountainous areas in the north and thinner in the lower-elevation Irtysh River valley (Figure 6a). The snow depth ranged from 16.23 cm to 34.19 cm in 2000–2019, with an average snow depth of 23.41 cm. The rate of thickening snow depth in the Chinese Altai Mountains was 0.27 cm a$^{-1}$ (Figure 6b). Based on the snow density data measured by Zhong et al. (2021) [33] and data from 212 ground snow sampling measurement sites and Jimnai County, located in the Sawir Mountains, maximum, minimum, and average snow density values of 0.28 g cm$^{-3}$, 0.11 g cm$^{-3}$, and 0.21 g cm$^{-3}$ were obtained. Based on the assumption that snow density is spatially uniform and constant throughout the season, combined with snow cover area and snow depth, the maximum, minimum, and average snow water equivalents in the Chinese Altai Mountains were 1.49 km$^3$, 0.58 km$^3$, and 1.12 km$^3$, respectively. Table 1 shows the snow cover area, snow depth, and snow cover water equivalent of each basin in the Chinese Altai Mountains. It can be found that the maximum snow cover area, snow depth, and snow water equivalent occurred in the Burqin River Basin. The minimum snow cover area, snow depth, and snow water equivalent occurred in the Lhaster River Basin. The spatial distribution of snow is influenced by elevation and topography. Snow cover area and snow depth were greater in the high-altitude and alpine regions, and thinner snow cover was observed in the sliding plain. This result was consistent with the findings by Zhong et al. (2021) [33].

**Table 1.** Distribution of snow cover parameters in the basins of the Chinese Altai Mountains from 2000 to 2022.

| Parameters | Period | Haba | Burqin | Kelan | Karaertis | Kayit | Lhaster | Chinese Altai Mountains |
|---|---|---|---|---|---|---|---|---|
| Basin area (×10$^4$ km$^2$) | - | 0.77 | 1.05 | 0.97 | 0.64 | 0.73 | 0.94 | 5.1 |
| Mean snow cover area (×10$^4$ km$^2$) | 2000–2022 | 0.28 | 0.61 | 0.37 | 0.32 | 0.4 | 0.29 | 2.27 |
| SD_mean (cm) | 2000–2019 | 27.23 | 35.6 | 21.16 | 20.03 | 19.93 | 7.66 | 23.41 |
| SWE_min (km$^3$) | 2000–2022 | 0.08 | 0.24 | 0.09 | 0.07 | 0.09 | 0.02 | 0.58 |
| SWE_max (km$^3$) | 2000–2022 | 0.21 | 0.61 | 0.22 | 0.18 | 0.22 | 0.06 | 1.49 |
| SWE_mean (km$^3$) | 2000–2022 | 0.16 | 0.46 | 0.16 | 0.13 | 0.17 | 0.05 | 1.12 |

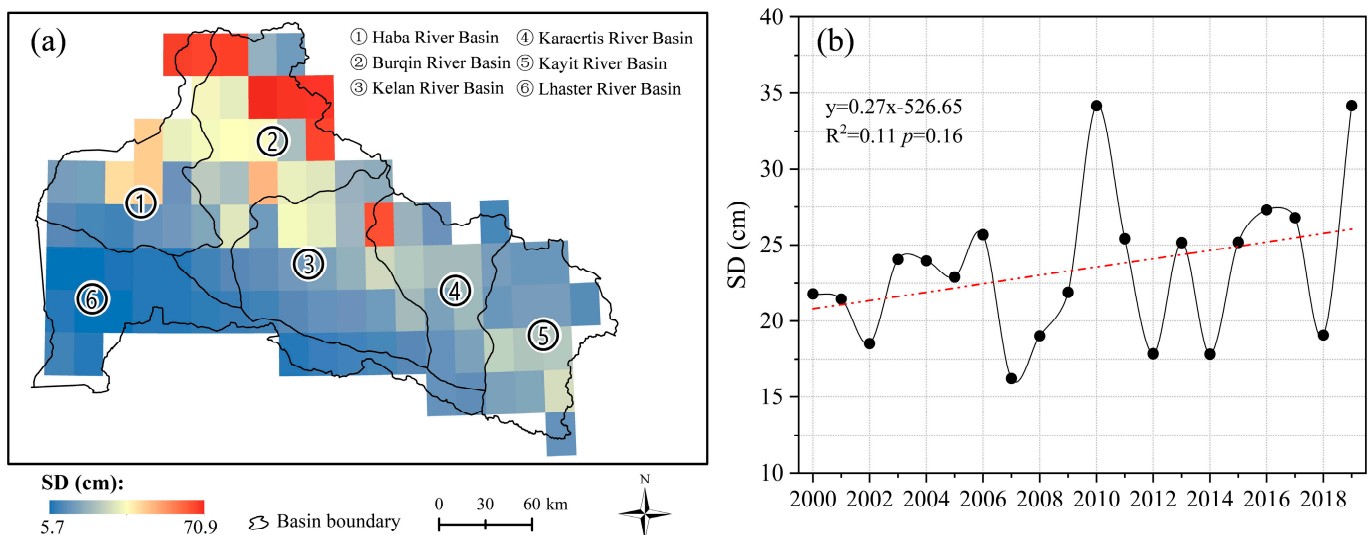

**Figure 6.** Spatiotemporal variation of snow depth in the Chinese Altai Mountains from 2000 to 2022. Spatial pattern (**a**) and interannual variation (**b**) of snow depth in the Chinese Altai Mountains from 2000 to 2022.

## 5. Discussion

### 5.1. The Influence of Climatic Change on the SCF

Regional SCF is affected by abnormal thermal conditions in the atmosphere, changes in meteorological parameters, and direct coupling between the surface and atmosphere. Based on the interannual variation sequence of SCF from 2000 to 2022, using higher or lower than the difference between the average state and one standard deviation as a classification criterion, we ultimately selected three years with higher SCF in 2001, 2010, and 2021 and three years with lower SCF in 2000, 2007, and 2020. Figure 7 depicts the 500 hPa anomalies of wind and divergence in the years with high and low SCF over the Chinese Altai Mountains, which were affected by westerlies and polar air mass throughout the year.

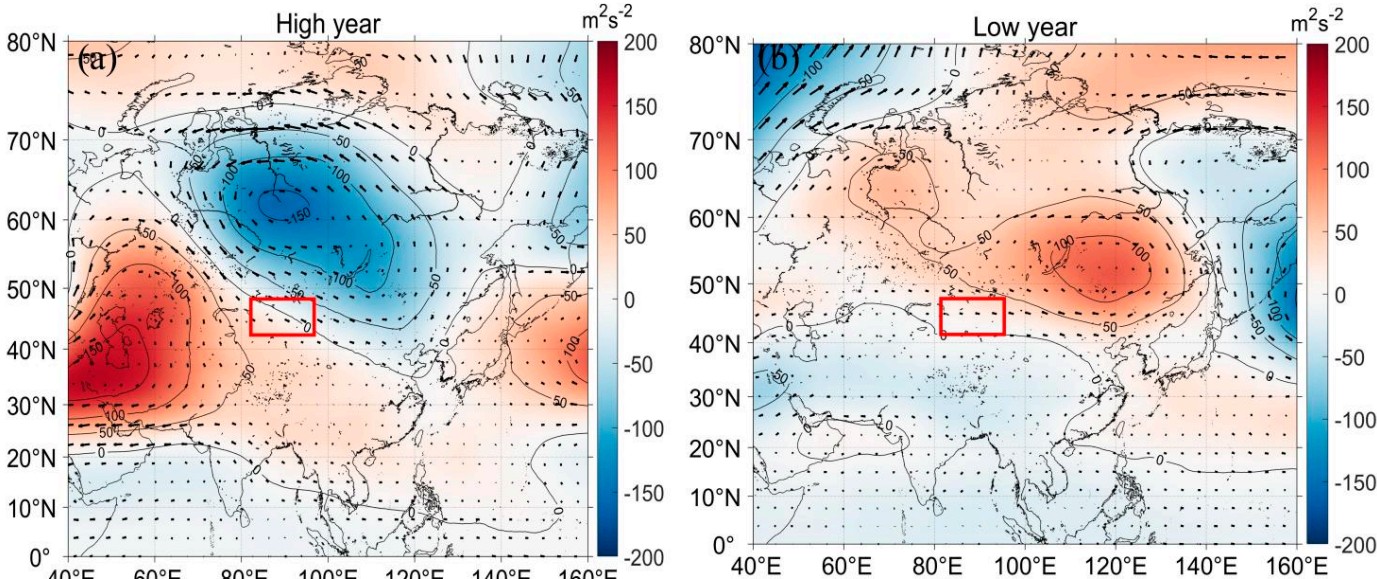

**Figure 7.** Composite anomalies of the 500 hPa wind and geopotential height in the years with high SCF (**a**) and low (**b**) SCF. The red box indicates the Chinese Altai Mountains.

The air masses that affect precipitation in the Chinese Altai Mountains include the sea source transported by the westerly wind, the water vapor replenished along the way,

the water vapor in the polar Arctic Ocean, and the near-source water vapor of local re-evaporation. From the perspective of wind fields in the lower troposphere, the regions in the Chinese Altai Mountains were mainly controlled by anomalous westerly airflow in years with high SCF (Figure 7a). The "Anticyclone–cyclone" wave train from the tropical West Indian Ocean to the west of Lake Baikal via the Arabian Sea transports the tropical warm and humid air to the middle and high latitudes. The anomalously low trough of the height field near Lake Baikal is conducive to guiding cold air southward. Anomalous westerly winds and northwest and southwest airflows converge over the Chinese Altai Mountains, which is conducive to the convergence of cold and warm airflows to produce snowfall. In years with low SCF (Figure 7b), the geographical height of the Arctic region was below the average state, indicating that cold air may be difficult to transport out of the Arctic region. The anomalous anticyclonic circulation near Lake Baikal caused the regions over the Chinese Altai Mountains to be controlled by anomalous easterly and northeasterly winds. Anomalous easterly winds at high altitudes and insufficient northward transport of water vapor at low latitudes are not conducive to snowfall. However, the SCF in the north of the Chinese Altai Mountains was higher than that in the south, regardless of the years with high or low SCF. This is usually because the polar air mass carries less water vapor and, with the blocking of mountains, its impact on the precipitation in the south of the Chinese Altai Mountains is relatively small; thus, the northern part of the Chinese Altai Mountains may be more affected by the polar water vapor.

The bivariate relationships between climate variables and SCF in the Chinese Altai Mountains for seasonal variation are summarized in Table 2. Generally, there were different correlations between SCF and air temperature/precipitation in different seasons because of the diversity in precipitation/air temperature patterns, precipitation amounts, and seasonal snow-melting rates. On the annual time scale, the regions with negative correlation between air temperature and SCF (Figure 8a,b) accounted for 56.03% of the territory of the Chinese Altai Mountains, of which 53.27% did not have statistically significant negative variations. Regions with significant negative correlations (including $p < 0.01$ and $p < 0.05$) were distributed along the Irtysh River. However, the regions with significant positive correlations only accounted for 0.71% of the total area, indicating that the negative correlation between SCF and temperature was dominant in the Chinese Altai Mountains. The regions with no statistically significant negative variations between precipitation and SCF accounted for 97.76% of the territory of the Chinese Altai Mountains (Figure 8c,d). Therefore, the SCF variation in the Chinese Altai Mountains from 2000 to 2022 was mainly controlled by air temperature instead of precipitation.

In terms of seasonal variation, the main driving factor of SCF change was the air temperature in spring (Figure 8f,h). Regions with negative correlations accounted for 32.87% of the total area, considering a risk of error of less than 1%, and 21.62% of the total area, considering a risk of error of less than 5%. In summer, the regions with significant correlations between SCF and air temperature/precipitation were scattered, accounting for 2.59% and 3.49% of the territory of the Chinese Altai Mountains (Figure 8l). SCF variations were affected by the combined effects of air temperature and precipitation during this season. In autumn, regions with significant negative correlations between SCF and air temperature accounted for 32.87% of the total area, of which 17.37% of the total areas ($p < 0.01$) were distributed mainly at lower elevations (Figure 8m,n). Regions with significant positive correlations between SCF and precipitation accounted for 7.8% of the total area (Figure 8o,p). Air temperature was once again converted to the main meteorological parameter controlling the changes in SCF. In winter, because precipitation often occurred in the form of snowfall, the impact of air temperature on snow cover was weakened, resulting in the increased stability of snow cover and the maximum SCF value. Even though there was a decreasing trend in precipitation, the SCF values maintained an upward trend. Therefore, there was no significant correlation between SCF and air temperature (Figure 8q,r), but it was negatively correlated with precipitation during the study period (Figure 8t, 12.14%). In summary, there was no significant correlation between

SCF and temperature or precipitation in the high mountains of the Chinese Altai Mountains at any timescale. Considering a risk of error of less than 5%, the regions with strong correlations were mainly distributed at low elevations in the Chinese Altai Mountains, near the Irtysh River. This phenomenon may be related to the sporadic distribution of seasonal snow cover or changes in seasonal snow cover, with air temperature and precipitation patterns at low elevations.

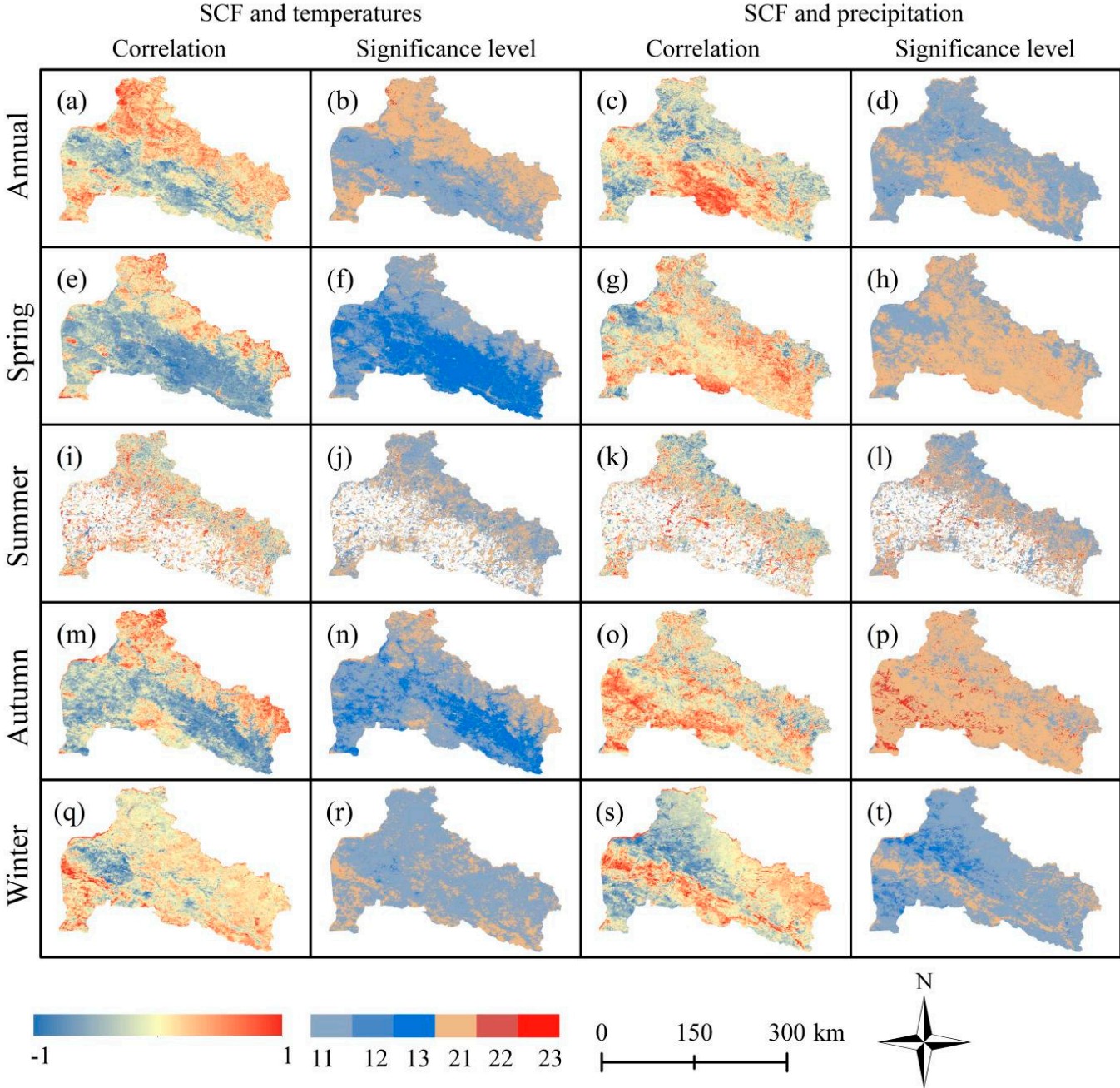

**Figure 8.** Statistical significance of correlations between SCF and precipitation and air temperature in the Chinese Altai Mountains from 2000 to 2022. Among them, correlation (**a,e,i,m,q**/**c,g,k,o,s**) and the significant correlation with statistical significance (**b,f,j,n,r**/**d,h,l,p,t**) between SCF and temperature/precipitation in the four seasons were shown.

**Table 2.** The area proportion (%) of correlation between SCF and temperature/precipitation with a statistical significance.

| Period | SCF and Temperature | | | | | | SCF and Precipitation | | | | | |
|---|---|---|---|---|---|---|---|---|---|---|---|---|
| | Positive Correlation | | | Negative Correlation | | | Positive Correlation | | | Negative Correlation | | |
| | $p < 0.01$ | $p < 0.05$ | $p > 0.05$ | $p < 0.01$ | $p < 0.05$ | $p > 0.05$ | $p < 0.01$ | $p < 0.05$ | $p > 0.05$ | $p < 0.01$ | $p < 0.05$ | $p > 0.05$ |
| Annual | 0.06 | 0.65 | 43.26 | 0.3 | 2.46 | 53.27 | 0 | 0.19 | 32.82 | 0.21 | 1.84 | 64.94 |
| Spring | 0.01 | 0.05 | 5.27 | 32.87 | 21.62 | 40.19 | 0.07 | 0.98 | 66.71 | 0.02 | 0.18 | 32.03 |
| Summer | 0.02 | 0.24 | 27.31 | 0.34 | 2.25 | 36.04 | 0.33 | 3.16 | 29.19 | 0.1 | 0.6 | 32.81 |
| Autumn | 0.06 | 0.22 | 15.31 | 15.5 | 17.37 | 51.54 | 1.36 | 6.44 | 78.51 | 0.01 | 0.07 | 13.61 |
| Winter | 0 | 0.04 | 21.04 | 0.08 | 0.63 | 78.21 | 0 | 0.03 | 14.39 | 2.75 | 9.39 | 73.44 |

### 5.2. Comparison with Previous Studies on Snow Cover Parameters

As the most stable snow-covered area in northern Xinjiang, changes in snow cover on the Chinese Altai Mountains is crucial to the sustainable use of regional water resources. In this study, the average annual SCF was 45%, decreasing at a rate of $1\%(10a)^{-1}$ from 2000 to 2022. The maximum in SCF occurred in winter and the minimum occurred in summer, which is consistent with the results by Qin et al. (2022) [48]. According to Zhao et al. (2021) [49], the average snow cover area in northern Xinjiang was approximately $25.6 \times 10^4$ km$^2$ from 1980 to 2019, decreasing at a rate of $0.41 \times 10^4$ km$^2 \cdot$a$^{-1}$. The snow cover area of the Chinese Altai Mountains was $2.27 \times 10^4$ km$^2$ from 2000 to 2022, occupying approximately 8.87% of the snow cover area in northern Xinjiang. According to in-situ observation data from meteorological stations, the average SD in the Chinese Altai Mountains was ~18 cm from 1980 to 2009, and the maximum SD reached 34 cm [28,50], which is consistent with this study, with an average SD of 23.4 cm.

In addition, we also comprehensively compared the changes in snow cover parameters in northern Xinjiang represented by the Chinese Altai Mountains, the Tibetan Plateau, and Northeast China (Figure 9). It has been found that there are significant differences in snow cover parameters among the three major stable snow cover regions based on existing observational data and previous studies [29,50–53]. In 2010–2019, the annual SCF in the Tibetan Plateau was ~15.7%~29.6% in spring, 5.4% in summer, 20.6% in autumn, and 17.5% in winter [54]. In 2002–2011, the SCF in Northeast China was relatively high in January and February, reaching 60% in winter [55]. From 1980 to 2020, the average snow cover area of the Tibetan Plateau was $63.2 \times 10^4$ km$^2$, considering a risk of error of less than 5%, an average reduction of $3.9 \times 10^4$ km$^2$ per decade [56]. The average snow cover area in Northeast China showed a weak but not statistically significant upward trend from 2002 to 2011 [55]. In 1951–2018, the average snow cover days in northern Xinjiang was ~80 d, increasing at a rate of ~0.6 d$\cdot$(10a)$^{-1}$. During the same period, the average snow cover days in Northeast China was ~77 d, with an increasing rate of ~2.3 d$\cdot$(10 a)$^{-1}$, and it was ~28 d, decreasing by ~0.5 d per decade in the Tibetan Plateau. The average SD in northern Xinjiang was ~8.6 cm and the maximum SD was ~16.9 cm. The overall daily SD was ~2.9 cm and the snow duration days was ~124 d. The average SD, maximum SD, overall daily SD, and snow duration days were ~5.8 cm, ~13.3 cm, ~1.7 cm, and ~141 d in Northeast China, respectively. Snow depth and snow duration days were ~2.9 cm, ~8.4 cm, ~0.3 cm, and ~168 d on the Tibetan Plateau. The snow onset/end dates in northern Xinjiang are similar to those in Northeast China, appearing in November and from the end of March to the beginning of April. The snow onset/end dates were earlier or later on the Tibetan Plateau, appearing in October and at the end of April, respectively. The snow onset/end dates for the three major stable snow regions showed varying degrees of delay or advance.

Overall, in the three major stable snow regions, the SD and snow cover days increased, whereas the snow duration days decreased. The distribution of snow cover is closely related to latitude, longitude, altitude, temperature, and precipitation [2,53,57]. Generally, the SD and snow cover area increase with increasing latitude, whereas the snow onset date, snow end date, and snow duration days are greatly affected by the interaction of altitude and air temperature. Therefore, in terms of single areas, the snow cover area and

snow duration days were the smallest in northern Xinjiang and the largest in the Tibetan Plateau. The SD, maximum SD, overall daily SD, and snow cover days were the largest in Northern Xinjiang and the smallest in the Tibetan Plateau (Figure 9). There is a typical temperate monsoon climate in Northeast China and a typical continental (inland) climate in northern Xinjiang. Geographical and climatic conditions are conducive to continuous snowfall in Northeast China and northern Xinjiang. Compared to northeastern China and northern Xinjiang, owing to its wide range, complex terrain, lower latitude, higher altitude, more precipitation, and higher air temperature, the characteristics observed in the Tibetan Plateau include an earlier snow onset date, later snow end date, longer snow duration day, fragmented snow distribution, and poor snow stability.

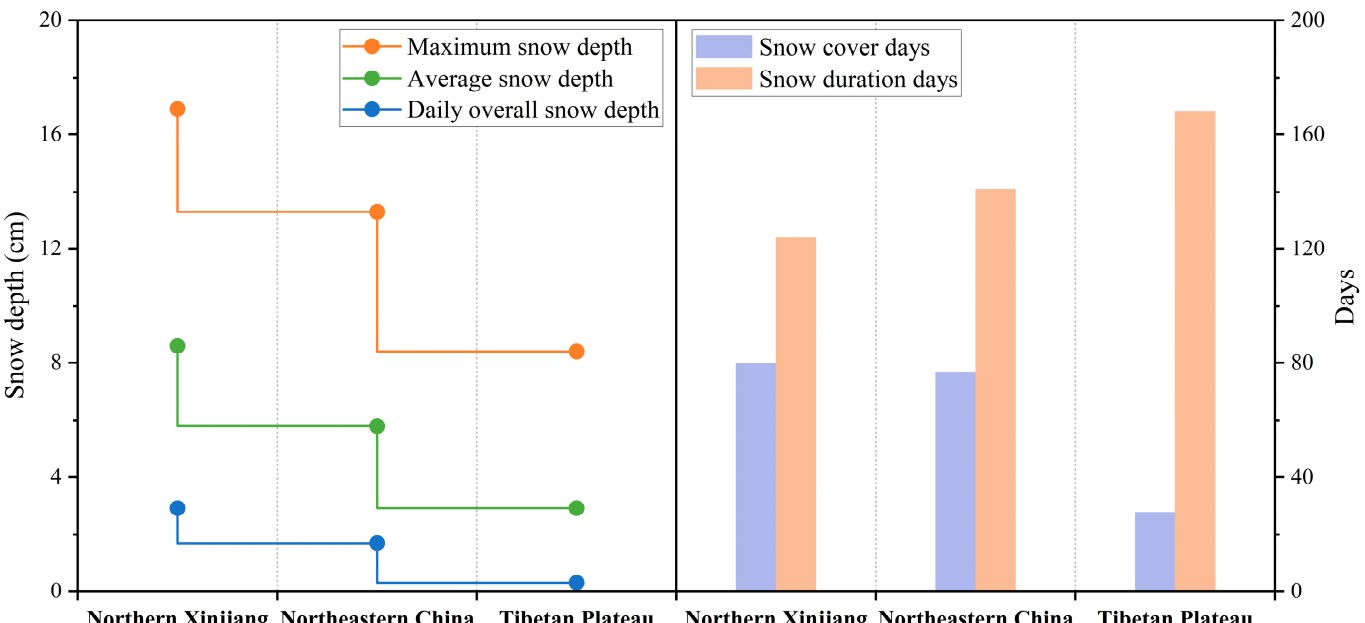

**Figure 9.** Comparison of snow cover parameters in the three major stable snow regions.

*5.3. Impact of Snow Meltwater on Regional Water Resources*

SWE refers to the depth of the accumulated water at which snow melts completely. Its variations change snow cover hydrology by regulating the total amount of water resources and reallocating their annual distribution. The runoff was approximately $100 \times 10^8$ m$^3$ in the Irtysh River Basin according to the report of Yang (1991) [58]. Based on the assumption that snow density is spatially uniform and constant throughout the season, the maximum, minimum, and average water equivalents of the Irtysh River Basin were estimated as ~$14.9 \times 10^8$ m$^3$, $5.8 \times 10^8$ m$^3$, and ~$11.2 \times 10^8$ m$^3$ in this study. From 2000 to 2022, the contribution of snow meltwater to annual runoff (snowmelt runoff ratios) ranged from 5.8% to 14.9%, with an average ratio of 11.2%. In addition, the sublimation and groundwater can lead to a decrease in snow water equivalent, but the estimated snow water equivalent in this study ignored the impact of sublimation and groundwater on snow cover. Previous studies have also reported the multi-year average runoff in some sub-basins of the Irtysh River. Wu et al. (2021) [35] reported an average annual runoff depth of 330.2 mm from 2000 to 2016 in the Kayit River Basin. From 1964 to 2011, Wu et al. (2022) [59] obtained an average annual runoff of $19.89 \times 10^8$ m$^3$ in the Haba River Basin, $5.44 \times 10^8$ m$^3$ in the Kelan River Basin, and $39.79 \times 10^8$ m$^3$ in the Burqin River Basin by hydrological year. Thus, the snowmelt runoff ratios were 25.30%, 8.04%, 30.22%, and 11.56% in the Haba, Kelan, and Burqin River Basins, respectively. However, the estimation of SWE introduces some uncertainties to this study owing to the inability to obtain detailed snow density data for each sub-basin. However, the results by Yang et al. (2022) [30] showed that the average annual snowmelt runoff ratios in Northern Xinjiang were more than 10% in the third-level basins and exceeded 30% in some basins, which indicates that the

discrepancy estimated for SWE was acceptable in this study. Since 2000, the annual runoff depth, snowmelt runoff, and snowmelt runoff ratios have increased marginally in the Kayit River Basin, which is consistent with the change in the Kelan River Basin from 1959 to 2005 [8,35]. In addition, as the main elements of the terrestrial cryosphere, glaciers and snow cover affect hydrology, water resources, and the water cycle on a regional or global scale. The contribution of glacial meltwater to runoff was 3.38% in the Irtysh River Basin in 1961–2006 and 3.4% in 1985–2015, according to studies by Gao et al. (2010) [60] and Liu. (2021) [61], respectively. As one of the key elements controlling regional hydrology and water resources, snow meltwater dominates the variations in surface runoff in the Chinese Altai Mountains, which is consistent with other studies [8,34,35]. The snowmelt runoff ratio is usually approximately 10% in Northeast China, and the contribution of snow meltwater to regional water resources is less than that in the Chinese Altai Mountains. Unlike the Chinese Altai Mountains, The snowmelt runoff ratios are usually higher than 10% in the Tibetan Plateau and may continue to decrease in the future [4,30]. Therefore, focusing on the interaction between snow cover, hydrology, and climate not only provides a scientific reference for global change research but is also associated with maintaining agricultural and economic development in a specific region.

## 6. Conclusions

The variability in snow cover parameters is crucial for regional disaster prevention and the scientific management of water resources in river basins, especially in the context of global warming. Based on the MOD10A2 snow product, in-situ observations, fused SD data, and meteorological data, and considering various snow cover parameters such as SCF, SD, snow density, snow cover area, and SWE, the spatial-temporal pattern and driving factors of snow cover change in the Chinese Altai Mountains in 2000–2022, the main conclusions of this study are as follows.

The average annual SCF value in the Chinese Altai Mountains from 2000 to 2022 was approximately 45.03%, with an overall increase rate of approximately $1.4\% \cdot (10a)^{-1}$. More than 60% of annual SCF values were distributed at high elevations in the Chinese Altai Mountains and less than 40% of annual SCF values were distributed on both sides of the Irtysh River. The SCF values and snow cover evenness were highest in winter. In addition, the average snow cover area was $\sim 2.27 \times 10^4$ km$^2$, the average SD was ~23.4 cm, and the average snow density was ~0.21 g·cm$^{-3}$. The snow water equivalent ranged from 0.58 km$^3$ to 1.49 km$^3$, with an average of 1.12 km$^3$ in the Chinese Altai Mountains in 2000–2022. The maximum values of SCF, SD, snow cover area, and SWE were found in the Burqin River Basin, whereas the minimum values were found in the Lhaster River Basin.

In years with high SCF, abnormal westerly airflow is favorable for water vapor transport to the Chinese Altai Mountains, resulting in strong snowfall, and vice versa in years with low SCF. Atmospheric circulation also changes the regional SCF variability by controlling air temperature and precipitation. Air temperature and precipitation are key meteorological elements affecting SCF changes, and air temperature is the main driving factor in the Chinese Altai Mountains, especially in spring and autumn. In total, 54.49% and 32.83% of the total area showed a significant negative correlation between air temperature and SCF at the 95% confidence level.

The SD, overall daily SD, maximum SD, and snow cover days were the largest in northern Xinjiang and the smallest in the Tibetan Plateau, and vice versa for snow cover area and snow duration. From 2000 to 2022, the contribution of snow meltwater to annual runoff can be calculated as 11.2%, 25.30%, 8.04%, 30.22%, and 11.56% in the Irtysh River Basin, Kayit River Basin, Haba River Basin, Kelan River Basin, and Burqin River Basin. The high snowmelt runoff ratio is a significant feature of the mountainous watersheds in the Chinese Altai Mountains.

**Author Contributions:** Data curation, H.L. and Z.Z.; Writing—original draft preparation, F.Y.; Supervision, P.W. and L.L. All authors have read and agreed to the published version of the manuscript.

**Funding:** This research was jointly funded by the National Natural Science Foundation of China (42371148), the Third Xinjiang Scientific Expedition Program (2021xjkk0801), the Youth Innovation Promotion Association of the Chinese Academy of Sciences (Y2021110), and the State Key Laboratory of Cryospheric Science (SKLCS-ZZ-2023).

**Data Availability Statement:** The data supporting the findings of this study are available from the corresponding authors under reasonable requirements.

**Conflicts of Interest:** The authors declare no conflict of interest.

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
