# Peer review of "The Variability of Snow Cover and Its Contribution to Water Resources in the Chinese Altai Mountains from 2000 to 2022"

_remotesensing, doi:10.3390/rs15245765_

Round 1

Reviewer 1 Report

Comments and Suggestions for Authors

The focus of the manuscript is not much on remote sensing – the introduction mentions only that snow cover will be ‘investigated base[d] on MODIS [data]’ and the discussion relates only to snow hydrology. In fact data and methods section reveals that the use of remote sensing is simply confined to making use of off-the-shelf MODIS data product MOD10A2, though the characteristics of this product are not discussed. In fact, other remote sensing data products are also used to estimate snow depth, as mentioned in section 3.1.3., but this lacks detail. I think a more interesting paper could be presented about the data fusion methods and an assessment of the reliability of the MODIS data product

Section 3.1.3 is quite hard to unpick. The verb ‘collect’ seems to refer to both primary data collection by the authors, and also to the analysis of previously collected data by other authors. I would appreciate a clear distinction in this section of which data were collected by the authors, and also which variables were directly measured and which estimated statistically.

Methods 3.2.1. This section would be easier to follow if the overall purpose was explained before describing the process in detail. Why did you choose a ‘unified latitude and longitude projection’ (I suppose this is geographical or Plate Carrée projection)? Could you explain what are ‘MODIS Reprojection Tools’ and give a reference to them (unless they are your own work)? What is the significance of pixel values of 100 and 200?

Methods 3.2.2 and 3.2.3 are quite long expositions of what are generally very well-known statistical analysis techniques, and could be considerably shortened. On the other hand, the analytical techniques (regression and correlation) are both parametric methods. Do the data justify their use, or would nonparametric methods be preferable?

Methods 3.2.4: accuracy assessment of MODIS product. The accuracy metric is not defined here, and the statement that ‘the results show that the MOD10A2 product can be applied for snow monitoring in the Chinese Altai Mountains’ lacks detailed justification.

Results section 4.4. I think it would be helpful to see a map of the estimated snow depth variable. I am concerned by fig 5 and the way it is interpreted in the text and in table 1. The comment in the text to the effect that, when the snow depth is less than 60 cm, the correspondence between measured and estimated depth is good, does not seem to be borne out in fact. Almost all of the data points correspond to an average snow depth of less than 60 cm, and include quite a lot of cases where the estimated (fused) value is much higher than the in situ value. Table 1 presumably uses fused values of SD to estimate SWE per basin. Were these values accepted as correct, or were they corrected to estimate the in situ snow depth using the linear regression relation? Is it meaningful to fit a linear regression relationship to the data of fig 5 that allows for a non-zero intercept? And would it not make more sense to determine this relationship using only data points for which the fused estimated SD was less than 36 cm (the maximum observed in table 1)?

5.1 Discussion. It is hard to assess the authors’ choice of exceptional years without seeing how the estimated SCF varies from year to year. Please consider showing this as a graph. Did you de-trend the data for the long-term variation before selecting these anomalous years?

Minor issues:

159 ‘h23v04’ and ‘h24v04’ are tile references, not orbit numbers.

Fig 3(a). Please indicate zero slope on the colour scale. Fig 3(b) Please align the significance level key with the figure. The codes ‘11.. 23’ are not very intuitive, and the colours used for codes 12 and 13 are quite hard to distinguish.

Fig 4 (b) (e) (h) (k) I think this colour key is not the same as for fig 3 (a). Would it not be helpful to use the same one?

Table 2 is clumsy. The % symbol does not need to be included 57 times in this table!

Fig 8 does not convey information particularly effectively.

Comments on the Quality of English Language

English is generally perfectly clear, although there are a few obscure phrases and typographic errors. Including notational errors, I found these:

Please correct superscripts as for example in line 15, and lines 307-321.

37-38 ‘accelerated to decrease’ is not a clear expression.

Table 1. Please use correct multiplication symbols, not asterisks.

295 ‘aggregated’ has superfluous hyphen.

298 ‘density’ has superfluous hyphen.

311 ‘estimated’ has superfluous hyphen.

Fig 1. caption needs to explain the source of the ‘snow properties samples’ since they were not collected as part of the work described in this MS.

159 ‘h23v04’ and ‘h24v04’ are tile references, not orbit numbers.

200 ‘reflect the largest snow cover area’ – the meaning of this is not clear.

Author Response

Dear Editor and Reviewers,

We would like to thank you for your careful review. We have carefully revised the manuscript by taking the stylistic comments into consideration. It is highlighted by yellow-coloring in the revised version for easier tracking. Our point-by-point reply to each comment is provided. A marked-up manuscript version showing the changes is attached to our reply to the comments.

Once again, thank you for your consideration of our revised manuscript for publication in Remote Sensing.

Best regards,

Fengchen Yu et al.

Reviewer 2 Report

Comments and Suggestions for Authors

The part of the work on the MODIS data is well conducted. However the part on the SWE should be removed or reworked since it is based on the assumption that snow density is spatially uniform and constant throughout the season. Snow density evolves across time and space, from 0.1 to 0.3 which can change the results of this paper by a factor of three.

"Snow density was calculated using the SD and SWE for the snow pillow sites. The average in situ snow density is ~0.21 g·cm−3, which was used to calculate the SWE in the basins of the Chinese Altai Mountains"

writing syle: avoid useless text like "It is worth noting that"

The review of the scientific literature is focused on chinese studies, but there are other studies especially in Europe and the USA which are based on the same remote sensing data and approach. It would be useful to review these previous works because they show how to deal with the cloud cover. Although MOD10A2 is already a composite which mitigates the cloud problem, it can still contain cloudy pixels which may create a bias in the snow cover area. These pixels should be interpolated.

This paper gives insight into this issue and how to deal with it.

Parajka, J. and Blöschl, G.: Spatio-temporal combination of
MODIS images – potential for snow cover mapping, Water Re-
sour. Res, 44, W03406, doi:10.1029/2007WR006204, 2008.

Section 3.2.1. (Snow cover frequency calculation) should indicate the time period to compute the SCF maps. I guess it is a hydrological year?

Figure 2. Instead of a map consider plotting the SCF by elevation band like here: https://doi.org/10.5194/hess-19-2337-2015, 2015. This is useful to compare the snow climatology to other regions.

Same for Figure 4. It is really hard to get any information from these maps. Trends should be aggregated by elevation classes or by catchments.

Table 1 should mention what is the date or period of these estimates. Is it at a date near peak accumulation? Or maybe it is the maximum value over the entire period ? As it stands this table is not useful information The seasonal snow cover is not static. It can be zero in summer.

Figure 7 is difficult to read, maps are too small. I do not think it is useful for the reader. Consider aggregating to catchment scale and present the results in boxplots?

The runoff ratios are highly uncertain it si important to specify the assumption to compute them 1) density assumption 2) sublimation is neglected 3) groundwater too

I hope my comments will be useful. Now let's reduce our carbon emissions to save the beautiful snow of the Altai mountains.

Author Response

(The authors gave the same response as above.)

Round 2

Reviewer 1 Report

Comments and Suggestions for Authors

The authors have addressed most of the issues I identified in the previous version. It remains true that there is little that is novel in the way of remote sensing in the manuscript, but the use of MODIS data products has been clarified. I have a small number of points to make now:

L39 ‘especially in April…’ The sentence is somewhat malformed. Presumably the intended meaning is that the mean annual rate of decrease is 0.14, while the April rate is 0.29.

l127 (and check elsewhere). I think (10a-1) should be (10a)-1.

L168 ‘sonw’ → ‘snow’

l200 Modis Reprojection Tool(s) – the manuscript still does not cite the relevant authorship. I guess it is Dwyer and Schmidt 2006?

Fig 5(a). Strange to have what look like fitted polynomial functions or spline curves (they are not defined in the figure caption) to data that are essentially periodic. E.g. all curves have positive gradients between autumn and winter, yet all need to return to lower values in spring.

Author Response

(The authors gave the same response as above.)

Reviewer 2 Report

Comments and Suggestions for Authors

The authors have adressed most of my concerns except that Tab 1 should be clarified.

Tab 1. should indicate the time of the "Snow cover area (×104 km2) 2000-2022".  Which Day of Year ? Or is it an average? What is the difference with the "Fractional snow cover (%)" (same value divided by catchment area?)

Author Response

(The authors gave the same response as above.)
